# Understanding Interstitial Lung Diseases Associated with Connective Tissue Disease (CTD-ILD): Genetics, Cellular Pathophysiology, and Biologic Drivers

**DOI:** 10.3390/ijms24032405

**Published:** 2023-01-26

**Authors:** Giuliana Cerro Chiang, Tanyalak Parimon

**Affiliations:** 1Division of Pulmonary and Critical Care Medicine, Department of Medicine, Cedars-Sinai Medical Center, Los Angeles, CA 90048, USA; 2Women’s Guild Lung Institute, Cedars-Sinai Medical Center, Los Angeles, CA 90048, USA

**Keywords:** interstitial lung diseases, lung fibrosis, connective tissue diseases

## Abstract

Connective tissue disease-associated interstitial lung disease (CTD-ILD) is a collection of systemic autoimmune disorders resulting in lung interstitial abnormalities or lung fibrosis. CTD-ILD pathogenesis is not well characterized because of disease heterogeneity and lack of pre-clinical models. Some common risk factors are inter-related with idiopathic pulmonary fibrosis, an extensively studied fibrotic lung disease, which includes genetic abnormalities and environmental risk factors. The primary pathogenic mechanism is that these risk factors promote alveolar type II cell dysfunction triggering many downstream profibrotic pathways, including inflammatory cascades, leading to lung fibroblast proliferation and activation, causing abnormal lung remodeling and repairs that result in interstitial pathology and lung fibrosis. In CTD-ILD, dysregulation of regulator pathways in inflammation is a primary culprit. However, confirmatory studies are required. Understanding these pathogenetic mechanisms is necessary for developing and tailoring more targeted therapy and provides newly discovered disease biomarkers for early diagnosis, clinical monitoring, and disease prognostication. This review highlights the central CTD-ILD pathogenesis and biological drivers that facilitate the discovery of disease biomarkers.

## 1. Introduction

Interstitial lung diseases (ILDs) are a group of heterogeneous pulmonary disorders that primarily affect the interstitial compartment of the lung parenchyma. The principal abnormalities are characterized by a pathologic accumulation of inflammatory and mesenchymal cells [1,2]. These cellular changes lead to overproduction and abnormal deposition of extracellular matrix, primarily collagen, causing fibrosing ILD. There are approximately 199 cases of ILD per 100,000 people in the United States. The prevalence is higher in females (218.9 and 179.7 per 100,000 females and males, respectively) [3].

ILDs can be classified into five major categories based on the etiology: primary diseases-associated (sarcoidosis, eosinophilic pneumonia, lymphangioleiomyomatosis or primary alveolar proteinosis), environmental exposure-associated (pneumoconiosis and chronic hypersensitivity pneumonitis), drugs-, chemicals- or radiation-associated, connective tissue diseases (CTD)-associated ILD (rheumatoid arthritis (RA-ILD), systemic sclerosis (SSc-ILD), Sjogren’s disease, inflammatory myopathies, anti-synthetase syndrome, systemic lupus erythematosus (SLE), and mixed connective tissue disease (MCTD-ILD)) and the idiopathic interstitial pneumonias which include idiopathic pulmonary fibrosis (IPF) and nonspecific interstitial pneumonias (NSIP) [4,5]. Among all these, IPF is the most common form of ILD, followed by CTD-ILD. While IPF has been extensively studied, data in CTD-ILD are limited.

CTD-ILD is a collection of systemic autoimmune disorders associated with lung interstitial abnormalities or lung fibrosis. Systemic sclerosis is the most commonly associated ILD (34.7%), followed by rheumatoid arthritis (20%), overlap syndrome (13.3%), autoimmune myopathies (6.7%), and undifferentiated CTD (5.3%). Unlike IPF, which is more prevalent in men, CTD-ILD affects more women at a younger age [6]. The underlying pathogenesis of CTD-ILD and CTD-ILD/fibrosis is generally controversial and not well-characterized as in IPF [7]. Given some overlap mechanisms, one can extrapolate IPF pathogenesis to CTD-ILD and fibrosis. Nonetheless, it is widely believed that dysregulation of inflammatory pathways is the primary cellular mechanism that drives profibrotic cascades causing aberrant tissue remodeling and fibroproliferation [8]. Further understanding the pathogenic mechanisms of CTD-ILD is essential to improve survival by tailoring treatment alternatives to target more specific regulatory pathways. In this review, we focus on the genomics and cellular pathophysiology of CTD-ILD and the biological driver mechanisms that can bring about new approaches or targets for early diagnosis and treatment of this incurable condition.

## 2. Pathogenic Mechanisms of Connective Tissue Disease-Associated ILD

### 2.1. Genetic Abnormalities

The finding that multiple members of the same family could develop pulmonary fibrosis suggested that there could be a genetic component responsible for the disease, as observed in familial pulmonary fibrosis (FPF). While most genetic variants have been studied in IPF, most recently, some variants in other conditions, such as CTD-ILD and hypersensitivity pneumonitis, have also been identified [6]. The common and well-characterized variants associated with ILD are the *MUC5B* and *TOLLIP* polymorphisms and genes affecting leukocyte telomere length.

#### 2.1.1. The *MUC5B* Promoter Polymorphism

Mucin 5B is a highly glycosylated protein normally expressed by airway epithelial cells that contribute to mucus production and maintains immune homeostasis [9]. A single nuclear polymorphism (SNP) on chromosome 11 (rs35705950) associated with a gain of function variant has been associated with increased expression of *MUC5B* [10]. Mutations in *MUC5B* are associated with mucociliary dysfunction, retention of particles that may function as antigens, and disruption of repair and regeneration mechanisms in the alveolar tissue [10,11]. The presence of the *MUC5B* promoter variant has associated IPF (OR = 9.0), and it is estimated to account for 30% of the overall risk [12]. Additionally, the *MUC5B* variant is greater in patients with interstitial pneumonia with autoimmune features (IPAF) and in patients with RA-ILD [13], which was validated by another study [14]. This association was more specific in a UIP pattern in the imaging [15]. On the other hand, population studies revealed that there was no association with pulmonary fibrosis in patients with SSc-ILD, suggesting a different pathophysiology [16] which correlates with the fact that, in patients with SSc-ILD, the most common radiologic and histopathologic finding is nonspecific interstitial pneumonia (NSIP) rather than a UIP pattern. To this end, the role of *MUC5B* promotor variants in CTD-ILD requires further studies.

#### 2.1.2. The *TOLLIP* Gene

The Toll-interacting protein gene (*TOLLIP*) synthesizes an inhibitory adaptor protein that acts downstream from the toll-like receptors, mediators of the innate and adaptive immune response, including the IL-1 receptor signaling. Genome-wide association studies (GWAS) have found three *TOLLIP* SNPs associated with reduced *TOLLIP* expression and IPF susceptibility [17,18,19]. Another study revealed that the minor allele C in *TOLLIP* rs5742890 was associated with worse survival and disease progression in patients with pulmonary fibrosis [20].

The studies of *TOLLIP* in CTD-ILD are inconclusive. A cohort of patients with CTD-ILD in Japan found a significant difference in survival on T allele carriers than for non-carriers (100% vs. 73% *p* = 0.024) [18]. On the other hand, a large multicenter cohort found a similar distribution of MAF rs5743890 in patients with RA-ILD, SSc-ILD, and controls. Moreover, this variant had no association with disease progression (change in predicted forced vital capacity (FVC)) and transplant-free survival in the CTD-ILD group [13].

#### 2.1.3. Telomere Mutations

Mutations in the telomerase enzymes and the sheltering protein complex, which protects them from premature shortening, have been linked to epithelial cell senescence, premature aging, and impaired remodeling response [12,21]. The telomere shortening can be detected in lymphocytes, granulocytes, and alveolar epithelial cells [22]. The suggestion that short telomeres may be implicated in the pathophysiology of pulmonary fibrosis stemmed from observational studies in patients with dyskeratosis congenita, a syndrome associated with premature aging, bone marrow failure, liver disease, and pulmonary fibrosis [23].

The most common telomere variants occur in six telomere-related genes: the telomere reverse transcriptase (*TERT*), the telomerase RNA (*TERC*) genes, dyskerin pseudouridine synthase 1 (*DKC1*, regulator of telomere elongation helicase 1 (*RTEL1*), poly(A)-specific ribonuclease (*PARN*), and the TECT1-interacting nuclear factor (*TINF2*). Patients with *TERT* and *TERC* mutations are younger at the time of diagnosis of pulmonary fibrosis, accounting for 15–20% of the cases of FPF [24]. Other telomere-associated genes identified are *RTEL1*, *DKC1*, and *PARN*, responsible for telomere stabilization and *TINF2* and nuclear assembly factor 1 (*NAF1*) [25]. These account for approximately 1–3% of familial cases of pulmonary fibrosis [26].

Shortened leukocyte telomere length, regardless of the presence of mutations and after adjustment for age, has been seen in patients with ILD, including the subgroup with CTD-ILD with 41.4% of patients below the 10th percentile, compared to 8.7% in healthy controls [27]. A study on patients with systemic sclerosis also revealed an association between shorter peripheral blood leukocyte telomere length and the development of interstitial lung disease [28]. Another study revealed that even though patients with interstitial pneumonia with autoimmune features (IPAF) had longer leukocyte telomere lengths (LTL) compared to patients with idiopathic pulmonary fibrosis, particularly in patients with IPAF, shorter telomere lengths (LTL < 10th percentile) were associated with a faster lung function decline and worse transplant-free survival compared to patients with LTL > 10th percentile [13]. Lastly, in RA-ILD, there were some similarities in telomere gene mutations with FPF [29]. Together, it is conceivable that telomere mutation promotes ILD or lung fibrosis in CTD patients, especially in lung epithelial cells.

In summary, the role of genetic mutations as a disease biomarker or disease prognosis in CTD-ILD is underwhelming, partly due to the rarity of the diseases. The current databases have expanded substantially in anticipation of more meaningful genetic information in these disease populations in the near future.

## 3. Pathologic Cell Types in CTD-ILD

As previously described, multiple lung cell types are involved in lung fibrosis pathogenesis, particularly mesenchymal and alveolar type II epithelial cells [30,31,32,33]. Most of the studies were in IPF. These pathologic cells utilize several basic cellular mechanisms to mediate lung fibroproliferation that are overlapped in all ILD/fibrosis. However, in CTD-ILD, the immune-mediated processes are likely the primary mechanisms [34]. Here, we describe multiple cell types in idiopathic and non-idiopathic forms of lung fibrosis that are relevant or shared pathologic phenotypes with CTD-ILD.

### 3.1. Lung Epithelial Cells

#### 3.1.1. Alveolar Type II Epithelial Cells (AT2)

Aberrant repairing mechanisms secondary to persistent or abnormal AT2 injuries associated with environmental, genetic factors, or chemical-induced damage are one of the well-characterized pathogenic mechanisms of lung fibrosis, but mostly in IPF [31,35,36]. They endure profibrotic phenotypic characteristics such as cellular senescence or can be affected by many profibrotic signaling: developmental pathways (TGFB, Wnt, and SHH), ER stress, autophagy/mitophagy, apoptosis, etc. [31,37]. In SSc-ILD, the number of AT2 cells was similar to the controls [38], whereas AT2 cells were dramatically decreasing in IPF, suggesting that AT2 may not be a major cell type in CTD-ILD. Furthermore, their specific profibrotic roles in CTD-ILD have yet to be defined.

#### 3.1.2. Basal Cells

Basal cells are located adjacent to the basement membrane. They have secretory and proliferative capacities, act as progenitor cells, and are involved in lung remodeling [39]. In IPF, the presence of basal cells is suggestive of their pathologic roles. For instance, the detection of basal cells in bronchoalveolar lavages of patients with IPF is associated with poor prognosis [40]. Single-cell RNA sequencing data indicated that basal cells include multipotent and secretory subsets with the predominance of secretory subtype in IPF [41]. In CTD-ILD, the pathologic role of basal cells is unknown. Some reports suggested that accumulation of basaloid-like cells inhibited normal lung repairs and conversely facilitated persistent fibroblast activation [42] Aberrant basaloid cells have also been found in lung tissue samples from patients with severe SSc-ILD with a dramatic loss of AT1 cells, similar to what was found in the IPF [43,44]. A pathologic functional study is needed to confirm the profibrotic effects of basaloid cells in SSc-ILD.

### 3.2. Pathological Fibroblasts

Fibroblasts are the major mesenchymal cells contributing significantly to lung repairing processes. Although a definition and characterization of pathological fibroblasts are ongoingly studied, expanding these cells in response to abnormal AT2 injuries is a main pathological feature in lung fibrosis, especially IPF [45,46].

In CT-ILD, persistent activation of fibroblast and connective tissue growth factors results in increased deposition of type I and III collagens and dysregulation of metalloproteinases (MMP) with overproduction of the extracellular matrix, which is a characteristic feature of SSc-ILD [47]. Unlike healthy controls, fibroblasts of patients with SSc-ILD have an inverse response to inflammation; in this population, there was increased production of the anti-apoptotic protein B cell lymphoma-2 (Bcl-2) in response to interleukin-6 (IL-6), whereas healthy controls expressed a pro-apoptotic protein Bcl-2-associated X protein (BAX) [48]. This finding supports the theory of the “apoptosis paradox”, whereby these pathologic fibroblasts become resistant to apoptosis [49]. Transcriptomic profiles of lung explants using single-cell RNA sequencing revealed that, compared to healthy controls, the fibroblasts of SSc-ILD lungs underwent extensive phenotypic changes demonstrated by the heterogeneity of these cells [38]. The major populations are myofibroblasts, SPINT2^hi^ fibroblasts, and MFAP5^hi^ fibroblasts. The increasing number of myofibroblasts and profibrotic characteristics of SPINT2^hi^ and MFAP5^hi^ contributed to ILD and fibrosis development in these patients [38].

### 3.3. Immune Cells

Immune dysregulation is one of the major pathogenic mechanisms that promote fibrosis in IPF [50], but is controversial due to the partial response to immunosuppressive treatment. However, in CTD-ILD, immune-mediated processes are the classical characterization of the basic pathogenesis given their underlying autoimmune-mediated conditions [7]. Many immunosuppressive agents are therapeutic cornerstones of CTD-ILD despite their limited efficacy. The principal immuno-regulatory pathways involve both innate and adaptive immune responses. The role of innate and adaptive immunity and associated cytokines, chemokines, and their effect in the interstitial space in the pathogenesis of CTD-ILD is detailed in Figure 1**.**

#### 3.3.1. Innate Immunity

##### Macrophages

Similar to fibroblasts, a heterogeneity of macrophages is involved in ILD or lung fibrosis. These subpopulations include alveolar and interstitium macrophages, monocyte-derived macrophages, and bone marrow-derived macrophages. Macrophages have profibrotic roles in lung fibrosis, well-characterized in IPF but not much in CTD-ILD. A lung microarray study on SSc-ILD patients indicated an accumulation of “activated macrophages” that was associated with progressive fibrosis [51]. We proposed that these functions may be plausible to CTD-ILD pathogenesis. For instance, a pre-clinical study suggests that alveolar cell injury activates inflammation and monocyte-derived macrophages, which drive lung fibrosis [52]. Macrophage proliferation is stimulated by macrophage colony-stimulating factor (M-CSF) and granulocyte-macrophage colony-stimulating factor (GM-CSF). Particularly, lung samples of patients with IPF showed different macrophage population proportions compared to normal lungs. Another study identified an aberrant profibrotic IPF macrophage subtype in patients with IPF as opposed to predominant inflammatory macrophages in the healthy controls [53]. In pulmonary fibrosis, macrophages release reactive oxygen species, cytokines, plasminogen activator, chemokines, growth factors, mitogens for mesenchymal cells, and leukotrienes, suggesting an augmented inflammatory capacity of alveolar macrophages in ILD [54,55]. Further studies showed that these macrophages share alveolar and interstitial macrophage characteristics and have pro-fibrotic immune cell functions [53]. Moreover, in another study, macrophage populations were also distinct at different stages of the disease, emphasizing the heterogeneity of this population [56].

##### Neutrophils

Neutrophils have been linked to pulmonary fibrosis and abnormal lung repair by releasing proteases, oxidants, cytokines, and chemokines affecting the extracellular matrix [57]. When compared to healthy controls, an increased neutrophil-to-lymphocyte ratio was found in patients with CTD-ILD and idiopathic pulmonary fibrosis, suggesting a role of neutrophils in their pathogenesis [58]. Additionally, neutrophil extracellular traps (NET) is a defense mechanism against pathogens. Activated by microbes, neutrophils can release DNA, histones, and antimicrobial peptides to form NETs to trap and kill microbes. However, in certain situations, they can lead to the formation of autoantibodies, cause direct injury to epithelial cells and result in an increased production of pro-inflammatory cytokines that induce the further formation of NETs, thus perpetuating the damage [59]. The enhanced NET formation has been found in patients with systemic lupus erythematous, rheumatoid arthritis, small cell vasculitis, and dermato and polymyositis [57,59,60] Particularly, when studied in pulmonary fibrosis, in vitro, NETs have shown to activate lung fibroblasts and contribute to their differentiation into myofibroblasts [61] which are key cells in the pathogenesis of CDL-ILD.

#### 3.3.2. Adaptive Immunity

##### T Cells

T cell lymphocytes are heavily implicated in the pathophysiology of connective tissue diseases. Recently, several studies have supported the role of T cells in the pathophysiology of CTD-ILD. Pulmonary T lymphocytes may regulate fibrosis by cell surface interactions that lead to fibroblast activation and proliferation as well as increased deposition of collagen in the extracellular matrix. Surgical lung biopsies from patients with CTD-ILD have demonstrated increased T lymphocytes in the lung tissue and lymphoid aggregates. Additionally, bronchoalveolar lavages from patients with SSc-ILD, RA-ILD, and inflammatory myositis have an accumulation of T cells with a predominance of cytotoxic CD8+ T cells [62].

Systemic sclerosis is a T cell-mediated autoimmune disease. The anti-topoisomerase A antibody (ATA) has been associated with a higher risk of ILD in this patient population [14]. Recent studies suggest that specific ATA CD4+ T cells with a proinflammatory-Th17 phenotype were found in patients with SSc-ILD compared to healthy controls with an association with a decline in lung volumes [63]. The role of ATA was extrapolated to other CTDs, and there was an increased expression in patients with CTD-ILD, particularly systemic sclerosis and Sjogren’s disease, compared to healthy controls [64].

The role of T lymphocytes was also seen in inflammatory myositis. Particularly, ILD associated with anti-MDA5 myositis has a high mortality due to the rapid progression of parenchymal disease. A cohort study in patients with rapidly progressive ILD showed a decrease of blood lymphocytes with an increased CD4:CD8 ratio suggesting an increase in cytotoxic activity with accelerated cellular destruction, which could lead to fibrosis due to the need for extensive tissue repair [65].

Single-cell transcriptomic profiling of peripheral blood mononuclear cells (PBMC) in patients with the anti-synthetase syndrome (ASS) associated with ILD demonstrated upregulation of interferon responses to NK-cells, monocytes, T cells, and B cells [66]. The increase of effector CD8: naïve CD8 ratio and Th1, Th2, and Th17 cell differentiation signaling pathways were also enriched in T cells of ASS-ILD patients suggesting their roles in ILD development. Additionally, angiogenic T cells, a specific T cell subset that promotes endothelial repair, were found to be in lower quantities in circulating blood from patients with CTD-ILD [67]. Overall, these studies show increased activation of cytotoxic T lymphocytes over repair mechanisms.

##### B Cells

B cell lymphocytes may also play a key role in the pathogenesis of CTD-ILD. In SSc-ILD, studies have shown extensive B cell infiltration with alveolar macrophages becoming M2 polarized upon induction of IL-4 and IL-10. M2 macrophages secrete profibrotic cytokines (CCL22, PDGF-BB, and IL-6) [34] that drive interstitial abnormalities and fibrosis. IL-6 has been studied as a predictor of disease progression, although with conflicting results [68,69].

In RA-ILD, there is an increase in CD4 cells and follicular B cell hyperplasia in the lung [7]. Humoral mediators may also play a role in fibrogenesis. IL-13 and IL-17 promote the differentiation of fibroblasts into myofibroblasts and promote fibrosis, respectively. In some cases, autoantibodies may play a key role. In SSc-ILD, the presence of anti-topoisomerase I is associated with the presence of ILD. On the other hand, in myositis-associated-ILD, no correlation has been found between the progression of ILD and the presence of autoantibodies [34].

In summary, multiple lung cell types are implicated in CTD-ILD with significant overlapping with other types of lung fibrosis. Lung fibroblasts and immune cells are the principal drivers in most CTD-ILD. The precise mechanisms how these cells regulate lung fibroproliferation remains to be elucidated.

## 4. Cellular Mechanisms and Regulatory Pathways

Dysregulation of lung repair following any type of stimulation is the basic pathogenesis of ILD and lung fibrosis. Lung repair and regeneration are regulated by multiple major pathways, including developmental pathways such as TGF-β, Wnt, Sonic hedgehog (SHH), inflammatory pathways, cellular senescence, amongst others [70,71]. Abnormal activation of these pathways contributes to lung fibrosis. Since these pathways are intertwined and can work in synergistic or antagonistic manners, identifying targeted molecules in lung fibrosis becomes challenging. The current knowledge does not indicate any distinct pathways that regulate CTD-ILD to other types of lung fibrosis or IPF. In this review, we focus on the pathways shown to be significant in CTD-ILD/fibrosis.

### 4.1. Transforming Growth Factor-Beta (TGF-β) Signaling

TGF-β regulatory signaling, specifically the TGF-β1 isoform, is the central developmental pathway in lung fibrosis pathogenesis [72]. Specifically, TGF-β seems to be a common final pathway of many profibrotic signals. Current FDA-approved anti-fibrotic agents, Pirfenidone [73,74] and Nintedanib [75], and a combination of both [76] are being studied in CTD-ILD as targeting agents of the TGF-β downstream signaling molecules further implicating this signaling pathway in CTD-ILD and fibrosis. Therefore, despite minimal data, distinguishing TGF-β regulatory genes and proteins are a promising approach to alleviate CTD-ILD.

Extrapolation of the TGF-β pathological roles in other types of ILD and fibrosis, specifically IPF, allows researchers to identify specific regulatory molecules of this pathway in CTD-ILDs where current evidence is limited. For example, TGF-β regulated genes were upregulated in macrophages of SSc-ILD lung tissues suggesting that TGF-β was involved in macrophage activations in mediating SSc-ILD [51]. In RA-ILD presenting as UIP, Janus kinase 2/signal transducer and activator of transcription 3 (JAK2/STAT3) were identified as intermediary molecules of TGF-β in activating myofibroblasts promoting lung fibrosis [77].

### 4.2. Inflammation Regulatory Pathways

#### 4.2.1. Toll-like Receptor (TLR) Signaling

Messenger RNA (mRNA) of the profibrotic TLRs, such as TLR2 and TLR9, was found in bronchoalveolar lavages in patients with CTD-ILD implicating its involvement in the pathogenesis of CTD-ILD [7]. TLR8 was upregulated in lung tissue of SSc-PF suggesting its role in regulating lung fibrosis [78]. The specific cell types utilizing this pathway in CTD-ILD needs further study.

#### 4.2.2. cGAS-STING and Type I Interferon Signaling Pathway

A cyclic GMP–AMP synthase (cGAS)–stimulator of interferon genes (STING) pathway is a cytosolic DNA sensor pathway that mediates innate immune response through interferon type I genes [79,80]. It also regulates T cell responses in tumors. This regulatory pathway is important in inflammatory lung diseases, especially autoimmune-mediated conditions such as SLE [81,82]. Specifically, upregulation of cGAS genes and cGAMP protein levels were demonstrated in PBMC of SLE compared to healthy controls and RAs [83].

#### 4.2.3. JAK/STAT Pathway

Janus kinase/signal transducer and activator of transcription (JAK/STAT) are families of tyrosine kinases that regulate numerous molecules upon being activated when such cytokines/growth factors bind to their receptors and thus are important in signaling downstream signaling molecules. The major proinflammatory cytokines are IL-4 and IL-6. The regulatory roles of JAK/STAT signaling pathway are well characterized in many CTD such as rheumatoid arthritis [77,83] and systemic sclerosis [84]. Furthermore, there is evidence that inhibiting this pathway could alleviate RA-ILDs [85]. Similarly, multiple JAK-inhibitors, tofacitinib, baricitinib, upadacitinib, and filgotinib, provided therapeutic benefits in SSc-ILD [86]. More importantly, in a pre-clinical model of scleroderma, a combination of anti-fibrotic and JAK1/2 inhibitors improved skin and pulmonary involvement suggesting a potential benefit in humans [87]. Some ongoing clinical trials are exploring these benefits.

### 4.3. Apoptotic-Pyroptosis-Ferroptosis Signaling Pathway

Apoptotic signaling plays an important role in lung fibrosis with bidirectional effects in different cell types [88]. Increasing apoptosis of lung progenitors (AT2 cells) diminishes normal lung-repairing processes and promotes aberrant signaling regulation of the lung regeneration [37]. On the other hand, resistance to apoptosis in pathological fibroblasts also causes dysregulation of lung repair resulting in lung fibrosis [89]. In autoimmune-mediated lung diseases, this axis may play important roles due to an imbalance of immune homeostasis. Transcriptomic analysis from multiple public databases of CTD-ILD lung cells revealed the upregulation of several genes in apoptotic regulatory pathways [90]. However, the functionality of those genes requires additional evaluation. A recent study indicated that profibrotic fibroblasts of SSc lungs upregulated ferroptosis-related genes such as ferroptosis-resistant (GPX4 and NR4A1), pro-ferroptosis genes (NCOA4, SAT1), and pyroptosis drivers (CASP4 and GSDMD) [91]. Further studies are ongoing to identify cellular and molecular targets associated with cell death pathways.

### 4.4. Cellular Senescence

Cellular senescence is a hallmark of aging processes and is well-described in lung fibrosis, primarily IPF, as one of the vital cellular mechanisms [32,92]. Senescent features of any lung cell type contribute directly and indirectly to abnormal lung fibroproliferation [93,94]. Despite its significant roles, this mechanism is less characterized in CTD-ILDs. A study of transcriptomic lung cell databases among 52 SSc-ILD patients indicated that the p53-dependent cellular senescence signaling pathway on lung fibroblasts was the major regulatory molecule driving ILD pathology in SSc patients [95]. Moreover, upregulation of senescence proteins p16 and p21 were demonstrated in CTD-ILD lung tissues suggesting that senescence contributes to ILD development [96].

## 5. Biomarkers in CTD-ILD

The main challenge in the management of patients with CTD-ILD is determining which patients with an underlying CTD will develop pulmonary findings and, of those, which ones will be clinically significant and will benefit from treatment. Given that not all patients present with respiratory symptoms at the time of diagnosis, there is a need for ancillary testing to identify the development and progression of the disease and identify the patients who will benefit from treatment. The fact that connective tissue diseases are systemic diseases with various degrees of organ involvement and severity makes isolating a biomarker specific to lung disease challenging.

### 5.1. Blood

Blood biomarkers that correspond to disease activity are desirable, given the ease of clinical collection relative to other sites. Two protein candidates deserve special attention, Krebs von den Lungen and Surfactant protein-D, which have been studied in CTD-ILD with promising results [97].

#### 5.1.1. Krebs von den Lungen 6 (KL-6)

KL-6 Is a mucin-like glycoprotein expressed on AT2s. In vitro, it has profibrotic and anti-apoptotic effects on lung fibroblasts. Overall, in CTD-ILD, the serum KL-6 level correlated negatively with healthy controls and patients with pulmonary infection and positively with radiologic disease severity [98]. Particularly in SSc-ILD, there was a good correlation between KL-6 levels pulmonary function tests, particularly diffusion capacity, and radiologic findings [99,100]. It also correlated with the severity of parenchymal involvement in RA-ILD and sarcoidosis [101]. KL-6 has also been studied as a predictor for disease activity in inflammatory myositis with good correlation with pulmonary function tests as well as good performance as a prognostic factor before treatment initiation [102]. In RA-ILD, increased levels of serum KL-6 were associated with the presence of active pneumonitis as well as a positive correlation between KL-6 and reticular opacities on CT findings [96,103,104].

#### 5.1.2. SP-A and SP-D

Surfactant proteins A (SP-A) and D (SP-D) are lipoprotein complexes secreted by AT2s to decrease surface tension in the alveoli, preventing collapse. In patients with SSc-ILD, SP-D levels were higher compared to healthy controls [99]. However, the correlation with radiologic findings or pulmonary function tests has not been consistent, with some studies showing a negative correlation and some of them a weak positive correlation [105,106,107].

#### 5.1.3. Interleukin 6

Interleukin-6 (IL-6) is associated with macrophage activation. Increased serum levels of IL-6 are seen in patients with SSc compared to healthy individuals, particularly in the subgroup with diffuse disease [108]. Within patients with SSc-ILD, increased levels of IL-6 are independently associated with DLCO decline and mortality in patients with SSc-ILD within the first year of diagnosis [47,109]. The role of IL-6 in the pathogenesis of SSc-ILD is also supported by two trials that evaluated tocilizumab, an IL-6 inhibitor, in patients with SSc-ILD. In these studies, patients who received tocilizumab had a slower decline of FVC compared to the placebo [110,111]. In addition, IL-6 has been studied as a potential biomarker to determine treatment efficacy in patients receiving cyclophosphamide, where patients with disease stabilization had lower IL-6 levels than patients with active disease [112] suggesting a role in therapeutic monitoring or disease activity.

#### 5.1.4. Other Biomarkers

Some tumor markers were found to be elevated in CTD-ILDs. For instance, studies in carbohydrate antigen 19-9 (CA 19-9), carcinoembryonic antigen (CEA), and carbohydrate antigen 125 (CA-125) showed some correlation with the severity and progression of CT-ILDs [96,113]. A similar report of these markers was observed in 165 CT-ILD patients [114] and primary Sjogren’s syndrome-associated ILD [115]. It remains to be seen what role these proteins play in the pathophysiology of lung disease and whether biological mechanisms can support the statistical correlation. The more recent interesting biomarker is vitamin D in 85 CTD-ILD patients compared to IPF and healthy controls and revealed that patients with CTD-ILD had lower levels of vitamin D than controls. Furthermore, within this group, the median survival was lower in the group with lower serum vitamin D levels [116]. This concurred with a prior study indicating that vitamin D deficiency in CTD patients had a higher prevalence of CTD-ILD [117].

### 5.2. Bronchoalveolar Lavage Fluid

The ease of blood collection is the biggest advantage of evaluating disease biomarkers, however, they may not adequately reflect the pulmonary environment. On some occasions, obtaining organ-specific samples offers more condition-specific markers, and CTD-ILDs have no exception. Examining bronchoalveolar lavage (BAL) fluid cellular, genetic, and proteomic profiles provides diagnostic and prognostic qualification benefits for some ILDs [118]. However, due to the invasive nature of the procedure and potential confounding factors such as infection or aspiration, the utilization of BAL fluid profiles as biomarkers is limited and less popular.

In CTD-ILD, bronchoalveolar cellular profiles in patients with SSc-ILD did not correlate with progression. However, in SSc-ILD and inflammatory myositis, the presence of neutrophilia was associated with poor outcomes [119,120]. Through mass spectrometry, protein patterns in BALs have been studied in CTD-ILD, aiding in identifying potential diagnostic biomarkers and therapeutic targets. A small study revealed different expression patterns of over 100 proteins in patients with CTD-ILD, which differed from healthy controls. The functions associated with these proteins were signal transduction, post-translational modification, protein turnover, chaperones, secretion, amino acid and lipid transport, and metabolism [121]. In SSc-ILD, a small study evaluated the protein composition of BAL in patients with a UIP pattern and found increased expression of mannose receptor C1 (MRC1), associated with the activity of type 2 macrophages involved in tissue repair and fibrosis, as well as potential biomarkers, C3a, protein 12-3-3e, a regulator of surfactant-associated protein A1 (SPFA2) [122]. Cytokine levels could also function as biomarkers for the diagnosis of ILD. A study in 32 patients by SSc by Schmidt et al., found higher levels of interleukins 7, 4, 6, 8, and CCL2 in patients with lung involvement [123].

Together, there are several challenges to the discovery of biomarkers. Connective tissue diseases are heterogeneous, with a broad spectrum in clinical presentation and degree of organ involvement, and variable disease course. Moreover, most patients receive CTD-specific therapeutic agents that can alter disease-specific biological markers. Lastly, the variability of control groups can cause study biases. For instance, healthy and CTD without ILD control groups are likely provide different results.

## 6. Conclusions

Fibrotic ILD is a rare but mostly untreatable complication in several common CTDs, such as RA, scleroderma, SLE, idiopathic inflammatory myositis, and Sjogren’s syndrome. Significant histopathological overlap exists between CTD-ILD and IPF; therefore, it is conceivable that these two entities may share the underlying pathogenic mechanisms, and the information could be extrapolated from many IPF studies. Despite many similarities, the efficacy of anti-fibrotic or anti-inflammatory agents, or both, is limited in CTD-ILD, implicating potential unique regulatory pathways yet to be discovered. While many cell types contribute to IPF pathogenesis, the principal cellular players in CTD-ILD seem to be immune cells as the initiators and lung fibroblasts as the effectors. Anti-inflammatory agents are the mainstay treatment in CTD-ILD, especially during the acute phase, emphasizing the vital regulatory function of inflammation. However, those agents can only slow the progression of ILD. The preliminary data of anti-fibrotic or a combination of anti-inflammatory and anti-fibrotic treatment revealed the same results. Together, more specific targets or more accurate timing of treatment, or both, are crucial and urgently needed to improve CTD-ILD outcomes.

The critical challenges are the need for suitable pre-clinical models to interrogate clinical findings, potential therapeutic targeted molecules, and biomarkers for accurate prediction or early detection of ILD. Current scientific technology, such as spatial multi-omics, precision-cut lung slices (PCLS), and 3-D organoid culture systems, allow scientists to study pathogenic mechanisms directly from CTD-ILD human explanted lung specimens, with a limitation that these end-stage lung specimens provided limited values for early disease pathogenesis.

Additional clinical studies to evaluate the difference between CTD-ILD and other types of ILD/fibrosis would offer better insights. For instance, the difference of IL-8 in BAL among CTD-ILD, IPAF, and IIP, that CTD-ILD was the highest, suggesting a potentially different targeted pathway [124]. A multicenter analysis of this study will provide significant information. Moreover, screening biomarkers to identify high-risk CTD patients for ILD development is pivotal. Future studies should also focus on the differences between CTD without ILD and CTD-ILD, allowing better and earlier detection of patients who are more susceptible to ILD complications. Some specific genetic markers, as aforementioned, were helpful for screening, but this may be limited to a specific group of patients and still does not provide full therapeutic benefits. Overall, the uniqueness of the CTD-ILD population allows investigators to study longitudinal clinical courses of lung interstitial abnormalities and fibrosis from the pre-fibrotic stage to end-stage disease. More studies should be conducted to understand the underlying pathogenesis leading to interstitial and fibrotic lung diseases in these patients that may be extrapolative to other forms of ILD/fibrosis.

## Figures and Tables

**Figure 1 ijms-24-02405-f001:**
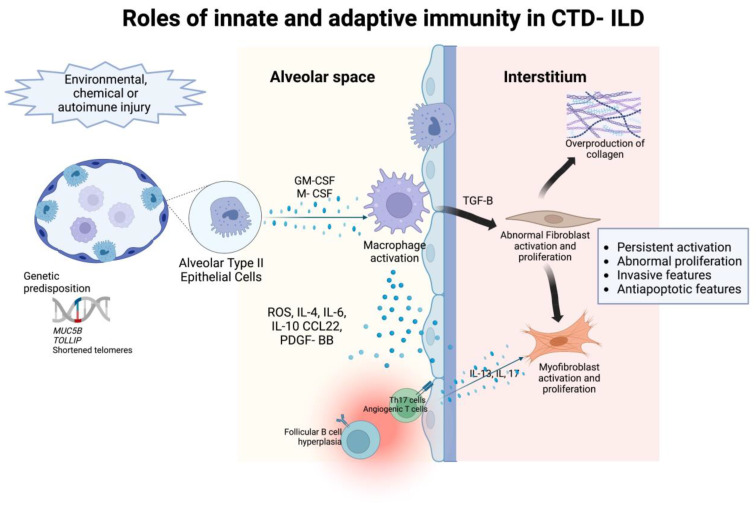
The roles of innate and adaptive immunity in the pathogenesis of intersitial abnormalities in connective tissue disease- associated interstitial lung disease.

## Data Availability

Not applicable.

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
