# Peer review of "Understanding Interstitial Lung Diseases Associated with Connective Tissue Disease (CTD-ILD): Genetics, Cellular Pathophysiology, and Biologic Drivers"

_ijms, 2023, doi:10.3390/ijms24032405_

Round 1

Reviewer 1 Report

It is an interesting paper covering pretty much the most of components involved into pathogenesis of ILD. Authors have done a really good job. A few minor comments only to be addressed prior to acceptance.

1 - More broad conclusions please. Please provide more insights regarding your thoughts about benefits of mass screening using high-throughput technologies in infants for detecting pre-disposition for ILD

2 - What if some of the mentioned aspects of pathogenesis can be utilised for diagnostic and/or genetic engineering purposes for overcoming current therapeutic hurdles of ILD. 

3 - Finally, your thoughts on MHC reprogramming as a way for ILD therapy in future ?

Reviewer 2 Report

Connective tissue disease-related interstitial lung disease is a difficult disease in clinical practice at present. Early diagnosis and effective treatment are the focus of current researchers. This research topic is relatively novel and absorbing. But here are some suggestions as follows.

1. Neutrophils are key innate immune cells, and their extracellular traps is also involved in the pathogenesis of this disease. Furthermore, the role of related innate immune cells in this disease should also be elaborated.

2. Innate immune signaling pathways include a variety of pathways, such as RIG-I, cGAS-STING signaling pathways, and interferon-related signaling pathways in the pathogenesis of this disease should not be ignored. Please search relevant databases and refine the role of relevant signaling pathways in this disease.

3. Are there related mechanisms or regulatory signals (Such as pyroptosis, necrotic apoptosis) involved? Please add it.

4. As a nonspecific inflammatory marker, the interleukin family were detected in a variety of CTDs and associated with multisystem involvement in CTDs. Therefore, they are not suitable biomarkers for diagnosing or predicting the occurrence of CTD-ILD. However, it may serve as a therapeutic monitoring indicator.

5. In conclusion, a vision for the future should be added. Given that current studies on diagnostic markers of CTD-ILD mainly focus on the differences between CTD-ILD and healthy controls. Future studies on biomarkers should focus on the differences between CTD and CTD-ILD, in order to better early screen CTD patients who are more prone to have a complication of ILD.
